

# No detectable changes in reproductive behaviour of *Caenorhabditis elegans* males after 97 generations under obligatory outcrossing

Weronika Antoł, Jagoda Byszko, Alicja Dyba, Joanna Palka, Wiesław Babik and Zofia Prokop

Instytut Nauk o Środowisku, Uniwersytet Jagielloński w Krakowie, Kraków, Poland

## ABSTRACT

In *Caenorhabditis elegans*, a species reproducing mostly *via* self-fertilization, numerous signatures of selfing syndrome are observed, including differences in reproductive behaviour compared to related obligatory outcrossing species. In this study we investigated the effect of nearly 100 generations of obligatory outcrossing on several characteristics of male reproductive behaviour. A genetically uniform ancestral population carrying a mutation changing the reproductive system to obligatory outcrossing was split into four independent populations. We predicted that the transition from the natural reproductive system, where males were extremely rare, to obligatory outcrossing, where males comprise 50% of the population and are necessary for reproduction, will increase the selection pressure on higher effectiveness of mating behaviour. Several characteristics of male mating behaviour during a 15 min interaction as well as copulation success were compared between the ancestral and evolved populations. No significant differences in male mating behaviour or fertilization success were detected between generations 1 and 97 of obligatory outcrossing populations. We found, however, that longer contact with females increased chances of successful copulation, although this effect did not differ between populations. We conclude that either selection acting on male mating behaviour has not been strong enough, or mutational input of new adaptive variants has not been sufficient to cause noticeable behavioural differences after 97 generations of evolution starting from genetically uniform population.

## INTRODUCTION

Self-fertilization (selfing) has evolved in numerous taxa of plants, fungi and animals, and is commonly associated with the so called selfing syndrome. First described in flowering plants, the syndrome is defined as a characteristic set of morphological and functional reproductive properties observed in most selfing species–in particular, degeneration of traits involved in outcrossing. In plants, it is typically manifested as decreased pollen number and reduced pollinator-attracting traits such as flower size, nectar and scent (*Shimizu & Tsuchimatsu, 2015*); in animals, as the reduction of mating- and

Corresponding author
Weronika Antoł,
weronika.m.antol@gmail.com

cross-fertilization related traits (*Cutter, 2008*). Characteristics of selfing syndrome may evolve as a simple consequence of relaxed selection leading to decline or loss of functions that were adaptive in outcrossing ancestors, or as an adaptation to self-reproductive life history (*Fierst et al., 2015*; *Shimizu & Tsuchimatsu, 2015*). Either way, selfing syndrome provides a compelling example of how shifts in reproductive systems can profoundly affect the evolution of morphological, physiological and behavioural traits. Such effects are studied predominantly by comparative analyses of species varying in reproductive system, inferring past events and processes from the distribution of traits on phylogenetic trees of extant taxa. Here, we attempted to study the evolutionary effect of a radical modification in the reproductive system in real time. We have used the nematode *Caenorhabditis elegans* as the model system.

In *C. elegans*, populations are composed almost exclusively of hermaphrodites, which reproduce primarily by selfing (they cannot fertilize other hermaphrodites due to the lack of a copulatory organ) and, very occasionally, by mating with males. The males, however, are extremely rare in this species both in the laboratory (0.1–0.2%; *Stewart & Phillips, 2002*) and in (at least the majority of) natural populations (*Andersen et al., 2012*). Sex in *C. elegans* is determined by the ratio of X chromosomes to autosomes: XX individuals are hermaphrodites, X0–males (*Hodgkin & Brenner, 1977*; *Hodgkin, 1987*; *Hunter & Wood, 1990*; *Chandler et al., 2011*). Males can be produced either as a result of outcrossing (50% of offspring) or non-disjunction of X chromosomes during selfing; the latter was suggested to be the main source of males (*Hodgkin, Horvitz & Brenner, 1979*; *Chasnov & Chow, 2002*). Because of the rarity of males, natural selection acting on male-specific traits is weak, and a number of selfing syndrome symptoms are observed. For example, compared to closely related outcrossing species, *C. elegans* males produce smaller sperm and are less successful at mating (*Chasnov et al., 2007*; *Garcia, LeBoeuf & Koo, 2007*; *Cutter, 2008*) Interestingly, specific levels of male mating (in)efficiency vary among strains (*e.g.*, *Hodgkin & Doniach, 1997*; *Bahrami & Zhang, 2013*; *Gimond et al., 2019*); at the extreme, in some strains the males are unable to mate at all due to mutation in *mab-23* gene (*Hodgkin & Doniach, 1997*; *Chasnov & Chow, 2002*). Similarly, variation among strains has been observed for other male traits such as the production of copulatory plugs (*Hodgkin & Doniach, 1997*, *Gimond et al., 2019*). Also hermaphrodite traits appear to be affected by selfing syndrome: in contrast to females of related species, hermaphrodite *C. elegans* do not respond to a factor produced by males which in females/hermaphrodites from other *Caenorhabditis* species induces immobilization during copulation (*Garcia, LeBoeuf & Koo, 2007*). Furthermore, hermaphrodites do not actively search for mates, are reluctant to mate in particular before they run out of their own sperm, and can even eject the already injected sperm. This reduced expression of reproductive traits is hypothesised to result from relaxed selection for the maintenance of these traits or from positive selection on self-fertilization traits (*Cutter, 2008*; *Cutter, Morran & Phillips, 2019*).

The natural reproductive system of *C. elegans* can be experimentally changed to dioecy, *i.e.*, obligatory outcrossing, through genetic manipulations (*Hodgkin & Brenner, 1977*; *Hodgkin, 1980*; *Doniach & Hodgkin, 1984*; *Schedl & Kimble, 1988*; see also Table I in *Anderson et al., 2010*; *Gray & Cutter, 2014*). In populations with obligatory outcrossing

(dioecy), because of the elevated frequency of males (ca. 50% of the population), selective pressure on male-specific traits should be restored and therefore one can expect to see the effect of selection on traits facilitating copulation and fertilization, i.a. in reproductive behaviour. Several studies suggest that increased frequency of outcrossing indeed imposes stronger selection on *Caenorhabditis* male traits. As mentioned above, males of frequently outcrossing *Caenorhabditis* species are characterized by larger sperm than males of predominantly hermaphroditic species (*LaMunyon & Ward, 1999*), suggesting that the evolution of larger sperm is a result of competition between males for the access to fertilization as larger sperm outcompetes the smaller. Indeed, this hypothesis was later supported by an experiment (*LaMunyon & Ward, 2002*) where *spe-8(hc53)* mutation, transforming mating system to obligatory outcrossing, was introduced into four *C. elegans* strains–after 60 generations of obligatory outcrossing nearly 20% increase in male sperm size was observed. Similar result was obtained by *Palopoli et al. (2015)* using different strains and mutations–after 30 generations of evolution under obligatory outcrossing, sperm size in males increased by 10–15%, while no increase was observed in males from control populations evolving under ancestral reproductive system. In two out of three populations evolving under obligatory outcrossing, males also showed increased sperm competitiveness compared to control populations. Moreover, the authors report a 4-fold increase of copulation time in obligatorily outcrossing, compared to control, populations after 60 generations of evolution.

The aim of the present study was to investigate whether males from populations transformed from almost exclusively selfing to obligatorily outcrossing will evolve changes in sexual behaviour and increased efficiency of fertilization. We used the most common laboratory strain of *C. elegans*, N2 Bristol, which we had chosen as a model in our research program (which this study was part of) for two main reasons. First, N2 has been extensively employed in research since 1970s and hence has undergone thousands of generations of laboratory adaptation (*Sterken et al., 2015*). We had expected that this would prevent the confounding effects of the adaptation to laboratory conditions occurring over the course of our evolutionary experiment–which is sometimes a problem in such studies (*Teotónio et al., 2017*). However, this particular expectation has failed us: in another series of experiments within our research program (cf. *Antoł et al., 2022*) we have found signatures of adaptation to laboratory condition. Second, the selfing syndrome symptoms in terms of dwindled male reproductive traits are strongly pronounced in N2, more so than in some other strains (*Hodgkin & Doniach, 1997*, *Bahrami & Zhang, 2013*, *Gimond et al., 2019*), thus providing ample potential for improvement by selection after introducing obligatory outcrossing.

We have compared mating behaviour after 97 generations of evolution under obligatory outcrossing to the behaviour of the ancestral population (directly after obligatory outcrossing was introduced). The ancestral population was characterized by almost no genetic variation. Such a setup resembles the situation in nature, where new populations tend to be founded by few individuals and overall, the populations harbor low genetic variation (*Andersen et al., 2012*; *Richaud et al., 2018*).

Male mating behaviour in *C. elegans* can be conceptually divided into the following steps: (1) mate-finding, (2) response, (3) turning, (4) vulva location, (5) spicule insertion, (6) sperm transfer, which are controlled by at least 28 identified genes (*Barr & Garcia, 2006*). According to *Stockley (1997)*, in many species increased duration of copulation is connected with competition between males. In our case, we did not have enough image resolution to be able to determine the moment of copulation, therefore the measured proxy was the total time males spent in contact with females (including touching, sliding around the female's body as well as the suspected copulation but the exact interval could not be detected). Our main predictions were that after 97 generations, the males will be able to find the mates more quickly than the ancestral males, maintain physical contact with them for a longer time and be more successful in fertilization. Moreover, we wanted to test if longer contact with females (as a postulated proxy for copulation duration) indeed increases the probability of fertilization success. An additional trait we analysed was tail-chasing behaviour, where a male reacts actively while touching his own body with his tail (which is a copulatory organ). The tail allows the male to detect a potential mate; the neural and genetic mechanism of this sensing by tail is well-studied (*Barr & Garcia, 2006*; *Hart, 2006*; *Sherlekar et al., 2013*). Tail-chasing behavior has been previously observed in one of the strains of *C. elegans* (*Gems & Riddle, 2000*), as well as in another predominantly selfing species, *C. briggsae* (*Garcia, LeBoeuf & Koo, 2007*). In *C. elegans*, it was also observed in response to extracellular vesicles (ECVs) released from ciliated sensory neurons of wild-type animals (*Wang et al., 2014*)–therefore, as the authors indicate, ECVs can be potential mate clues. It was also suggested that tail-chasing males start to express mating behaviour on their own bodies because of an inability to discriminate between 'self' and a potential mate (*Garcia, LeBoeuf & Koo, 2007*). The phenomenon of mistakes in trying to find a mate is known also outside of nematodes, *e.g.*, in amphibians, where males are often found *in amplexus* with objects different than conspecific females (*Serrano, Díaz-Ricaurte & Martins, 2022*), which is a potentially costly behaviour. We have analysed duration of this behaviour to test whether it affects reproductive success and whether males evolving under obligatory outcrossing will manifest this behaviour less frequently than their ancestors.

## MATERIALS AND METHODS

### Populations

The obligatorily outcrossing population was obtained from a highly isogenic wild-type *C. elegans* N2 strain population, derived by 20 generations of single-hermaphrodite transfer. The *fog-2(q71)* mutation from JK574 strain was introgressed into this isogenic population by 10 cycles of introgression followed by 10 generations of brother-sister inbreeding; for the detailed description of the introgression procedure, see *Plesnar-Bielak et al. (2017)*; the procedure was implemented following *Teotònio et al. (2012)*. The *fog-2(q71)* mutation blocks sperm production in hermaphrodites, transforming them into functional females, while male spermatogenesis remains unaffected (*Schedl & Kimble, 1988*). The reproductive system was therefore altered from almost exclusive selfing with very rare males to obligatory outcrossing with an approximately 1:1 sex ratio. This

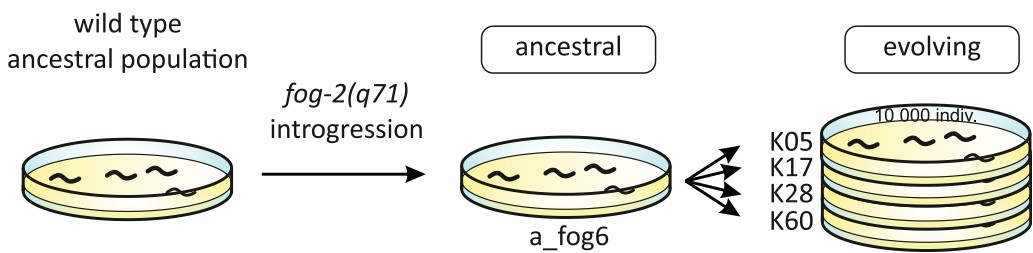

**Figure 1 Experimental populations.** Introgression of *fog-2(q71)* mutation to the wild-type ancestral population as well as further evolution of obligatory outcrossing ('fog') populations are shown.

ancestral population was then split into independently evolving replicates. Samples from the ancestral population were preserved by freezing at −80 °C for further comparisons. While the overall scope of our experimental evolution project was broader, including also populations with wild type reproductive system, the study reported here involved four evolutionary 'fog' populations derived from a single ancestral 'fog' population (Fig. 1). Wild type populations were not included since the study was strictly focused on the phenotypes expressed by males, which in the wild type N2 strain are vanishingly rare (∼0.01%) in our experimental wild type populations (personal observation).

The experimental populations were cultured as previously described in *Antoł et al. (2022)*, based on a standard procedure (*Stiernagle, 2006*; *Corsi, Wightman & Chalfie, 2015*). Briefly, populations of *ca*. 10,000 individuals were maintained at 20 °C on 14 cm Petri dishes filled with Nematode Growth Medium (NGM), covered with *Escherichia coli* OP50 strain as a food source. Every generation (*ca*. 4 days) the worms at L1-L2 developmental stage were transferred onto a fresh plate with bacteria. Every ∼12 generations samples of each population were frozen to enable further assays of phenotypes from different generations (nematodes can be propagated even after long-time freezing; *Brenner, 1974*).

## Experimental setup

We assayed the reproductive behaviour of a single ancestral population (a_fog6, Fig. 1) and four derived populations (K05, K17, K28 and K60, Fig. 1) at 97[th] generation of evolution under outcrossing. All populations were thawed from samples stored at −80 °C and allowed to recover for two generations before being used in the experiment. From the moment of thawing, the populations' names were encoded to hide the information about their identity so that the experimenters were not biased. The following experimental procedure was applied (Fig. 2). First, we isolated worms in the 4[th] larval stadium (L4), when the sexes are already distinguishable but the animals are still not capable of mating. For each population, we took eight L4 females and three L4 males and placed them on a fresh Petri dish with centrally located bacterial lawn (to facilitate them to gather in the central part of the plate), each sex separately: females on a 6 cm dish and males–on a 2.5 cm dish. Next day, when the animals matured, we placed one of the males on the Petri dish with females (outside of the bacterial spot) for 15 min and recorded his behaviour using a camera connected to binocular. We also observed the recording in real time,

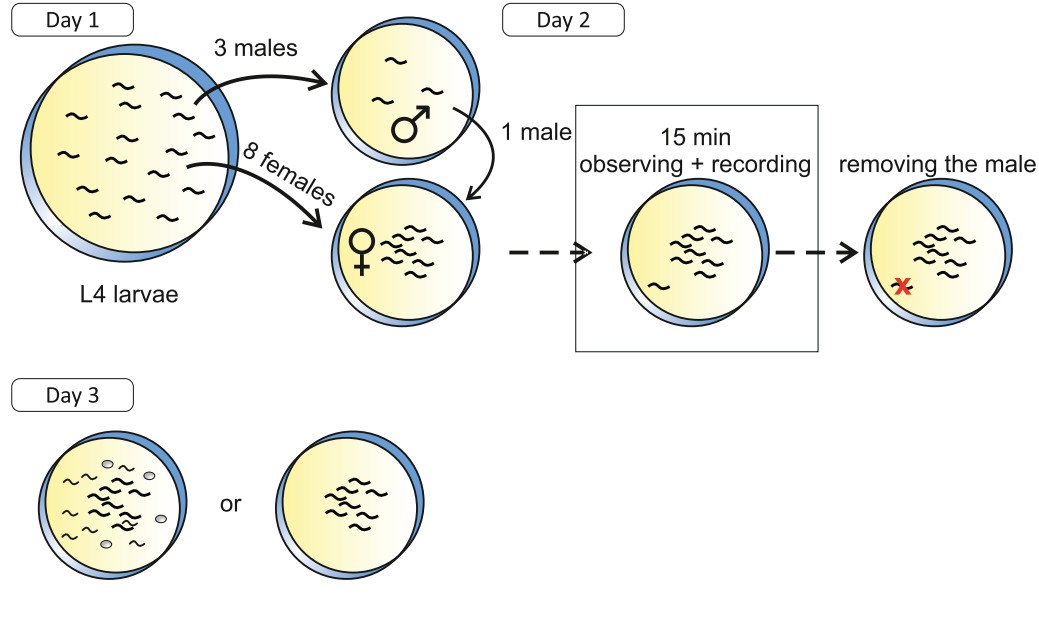

**Figure 2 Experimental setup.** The procedure presented here was performed in 10 replicates for each of five populations (ancestral + 4 evolved populations) in the experiment. On Day 1, L4 larvae were taken from a focal population: three males and eight females (each sex on a separate plate). On Day 2, when the animals matured, one male was placed on the plate with females and left there for 15 min. In this timeframe, his behaviour was noted and recorded (for detailed description of the traits observed, see main text); after that the male was removed from the plate. On Day 3, the offspring presence on the plate was checked and noted as a binary trait (0–no offspring, 1–offspring present).

adjusting the field of view so that the male was always visible, and we noted the following events: time to first contact of the male with any of the females (hereafter: time to first contact), duration of contact, time spent by the male chasing his tail. After 15 min, recording was stopped and the male was removed from the plate. On the next day, we checked the plates for the presence of eggs and scored this as a binary trait (0–no offspring, 1–offspring present). For each of the populations, the experimental procedure was planned to be performed in 10 replicates on 10 consecutive weekdays (one replicate from each of the five populations was performed each day). In case of one population, one replicate failed so that the total number of observations in the experiment was 49.

## Data analysis

Statistical analysis was performed using functions `lm` and `glm` from the R package stats (*R Core Team, 2022*). `Glm` was used for binary data (contact and offspring presence) with binomial error distribution and logit link function and `Anova` function from car package (*Fox & Weisberg, 2019*) was used to present the results as the overall effect of the factor (population) rather than contrasts between factor levels. The assumptions of the models were checked on diagnostic plots and, in the case of `glm` method, goodness of fit (control for overdispersion) was tested with `gof` function from `aods3` package (*Lesnoff, 2018*).

No substantial violations of the assumptions were detected. In all the analyses, time to first contact was set to maximum (900 s) in the cases where the contact did not occur within the observation period (the alternative procedure of removing such observations did not change the outcome of statistical analyses).

I. Analyses assessing the differences among populations in the following traits:

(1) Time to first contact analysed with the linear model.

```
model1<-lm (time_to_first_contact~population)
```

(2) Occurrence (or not) of at least one contact with any of the females (variable 'contact') was analysed with the general linear model with the binomial error distribution.

```
model2<-glm(contact~population, family="binomial")
```

(3) Total time spent in contact with females (variable 'contact_duration'), analysed with the linear model.

```
model3<-lm(contact_duration~population)
```

(4) Copulation success (presence of offspring the day after observation; variable 'offspring'), analysed with the general linear model with the binomial error distribution.

```
model4<-glm(offspring~population, family="binomial")
```

(5) Time spent by the male on chasing its tail, analysed with the linear model.

```
model5<-lm(tail_chasing~population)
```

In all the above analyses, the ancestral population was used as intercept so that all the populations from the 97th generation of evolution under outcrossing were compared to their ancestor.

II. Analyses of relationships between traits, with population included as an additional fixed factor:

(6) The effect of the time spent by the male on chasing its tail on the total duration of contact, analysed with the linear model.

```
model6<-lm(contact_duration~tail_chasing+population)
```

(7) The effect of the time spent by the male on chasing its tail on copulation success (presence of offspring), analysed with the general linear model.

```
model7<-glm(offspring~tail_chasing+population, family="binomial")
```

(8) The effect of the total duration of contact on copulation success, analysed with the general linear model.

```
model8<-glm(offspring~contact_duration+population,
family="binomial")
```

In models 6–8, the interaction between predictors was tested but in all cases turned out to be not significant so the final models were fitted without interaction.

## RESULTS

In total, 49 observations of the male behaviour and offspring presence were performed (10 for each of the tested populations except for one population from generation 97, K28, where only nine observations were available). The populations did not differ in time to first contact: the estimate for the ancestral population was 301.7 s, and those for the 97$^{th}$ generation: 208.6–414.1 s (Fig. 3A; $F_4 = 0.5813$, $p = 0.68$). In nine cases, no contact with female occurred during the entire 15 min observation period. There were no significant differences in frequency of no-contact observations between populations: two cases in the ancestral population, seven cases in all the evolved populations together (1/9 in K28, 3/10 in K05, 3/10 in K17) (Chi-square$_4$ = 6.02, $p = 0.20$). The populations did not differ in terms of the total duration of contact between the tested male and the females either: for the ancestral population, the estimate was 310 s, and those for the 97$^{th}$ generation: 124.9–317.4 s (Fig. 3B; $F_4 = 1.1007$, $p = 0.37$). Offspring appeared only in 12 cases: 2/10 replicates in the ancestral population and 10/39 replicates in generation 97 (1/10 in K05, 2/9 in K28, 3/10 in K60, 4/10 in K17); there were no significant differences in offspring presence between populations (Fig. 3C; Chi-square$_4$ = 2.83, $p = 0.59$). As for the time spent on tail-chasing, only one of the populations from generation 97 (K05) differed from the rest of the populations: estimated time 210 s ($t_4 = 2.956$, $p = 0.005$), while for the ancestral population the estimated time was 32 s and for the rest of the 97$^{th}$ generation populations this time was 8.3–105.89 s (Fig. 3D; $F_4 = 3.29$, $p = 0.02$).

The time spent on tail chasing did not affect either the total duration of contact with females (Fig. 3E; $F_1 = 0.9092$, $p = 0.35$) or offspring presence (Fig. 3G; Chi-square$_1$ = 1.9857, $p = 0.16$). The only variable that showed statistically significant relationship with offspring presence was the duration of contact with females (Fig. 3F; Chi-square$_1$ = 8.0732, $p = 0.0045$); however, its effect did not differ between populations (Fig. 3H; the interaction was not significant, neither was the effect of population).

## DISCUSSION

Our data did not confirm the prediction that 97 generations of evolution under obligatory outcrossing should increase male reproductive performance: we did not detect any differences between evolved populations and their ancestors in male sexual behaviours or the ability to fertilize females. These results are somewhat contrasting with earlier findings by *Palopoli et al. (2015)*, who observed that after 60 generations of experimental evolution under obligatory outcrossing (using the same *fog-2* mutation as in our experiment), duration of copulatory spicule insertion was four times longer than in control populations evolving under the ancestral reproductive type. However, our experiment differed from that of *Palopoli et al. (2015)* in two important aspects. First, we have compared reproductive traits between evolved and ancestral obligatorily outcrossing populations, whereas they compared males from obligatorily outcrossing (*fog-2* mutated) *vs.* predominantly selfing (*fog-2* wild type) evolved populations. Thus, the difference they observed in spicule insertion time could potentially be resulting directly from the *fog-2* mutation itself or linked mutations in neighboring genes, as well as from the subsequent evolution. Second, in the study by *Palopoli et al. (2015)*, the evolving lines were derived

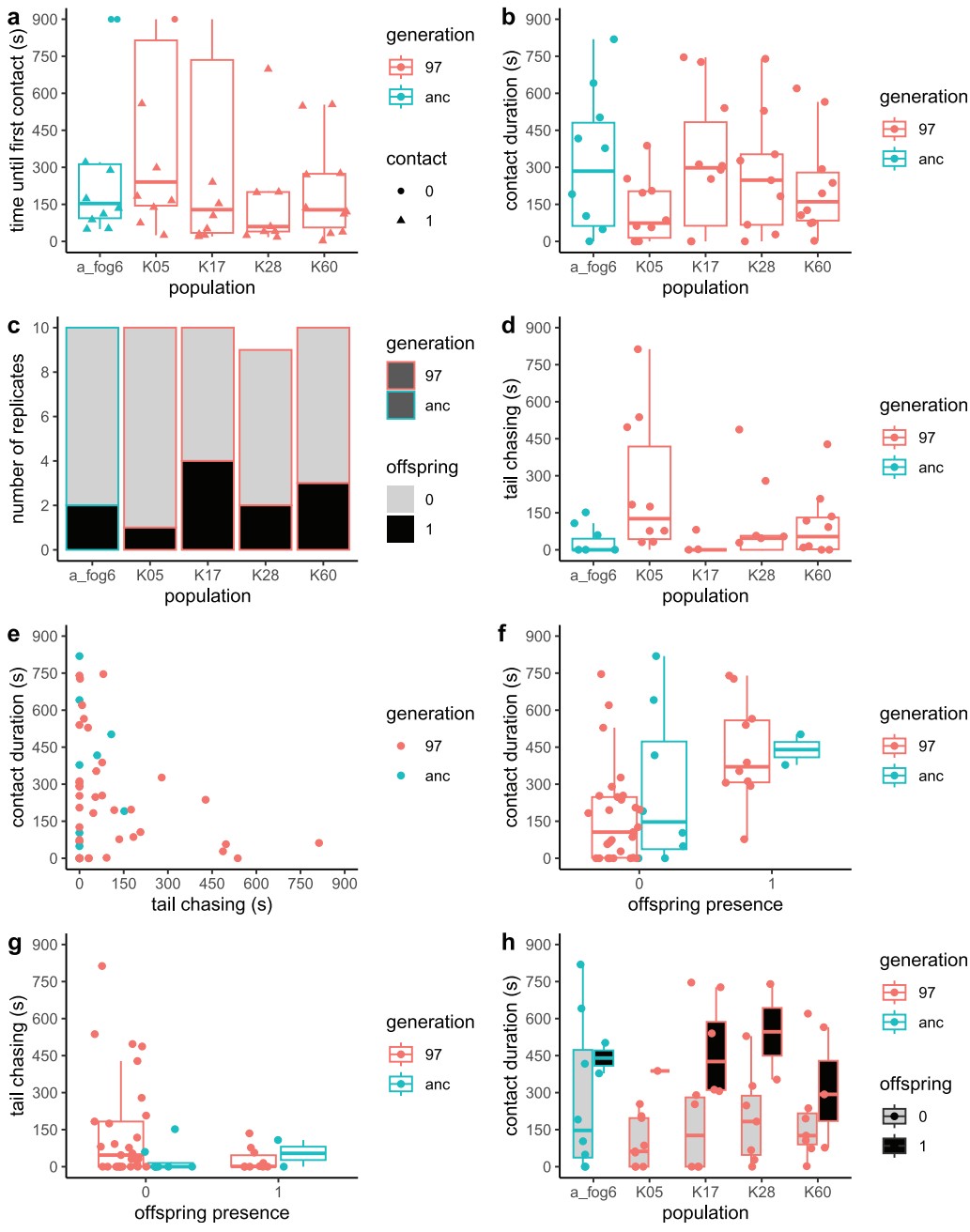

**Figure 3 Comparison of the ancestral and evolved populations.** In the box and whisker plots, bars represent median, the box encompasses the interquartile range (IQR: Q1-Q3) and whiskers extend to Q1-1.5*IQR and Q3+1.5*IQR. (A) Time to first contact with any of the females. (B) Total time spent in contact with females. (C) Offspring presence/absence. (D) Duration of tail-chasing behaviour. (E) The relationship between the total time spent in contact with females and the duration of tail-chasing behaviour. (F) Time spent in contact with females in populations where offspring was absent (0) or present (1). (G) Duration of tail-chasing behaviour in populations where offspring was absent (0) or present (1). (H) Total time spent in contact with females and offspring presence for each population.

from genetically variable ancestral populations, created by mixing 12 geographically distinct *C. elegans* isolates. Thus, the evolutionary changes in their lines most likely resulted from selection acting on the standing genetic variation available at the outset of the experiment. As the authors conclude, rapid changes they reported in several traits (as we have briefly summarized in the Introduction) indicate that many alleles underlying traits observed in naturally outcrossing *Caenorhabditis* species are available in *C. elegans* gene pool. In contrast to *Palopoli et al. (2015)*, here we were looking for evolutionary changes caused by selection acting on novel mutations, arising in populations initially devoid of genetic variation. Our results suggest that 97 generations were not sufficient for such changes to evolve.

Alternatively, our assays might have been unable to detect evolved behavioural changes due to differences in assay *vs*. experimental evolution conditions. During evolution, nematodes were kept at high density (ca. 10,000 individuals per 14 cm plate) and males had continuous access to females for many hours before being discarded during population transfer. In contrast, during behavioural assays density was much lower and time for sexual interactions was limited to 15 min. Thus, finding and inseminating females might have been more challenging for males in behavioural assays than under the conditions that they were evolving in. In an experiment by *Garcia, LeBoeuf & Koo (2007)* 10 min of interaction between males and females or hermaphrodites was sufficient to detect differences in reproductive behaviour between outcrossing and selfing *Caenorhabditis* species. However, the species assayed by *Garcia, LeBoeuf & Koo (2007)* diverged millions of years ago (*Fierst et al., 2015*) while our experimental populations had only 97 generations to accumulate differences so our study might be not sensitive enough to detect differences that have evolved within that time.

As described more broadly in the Introduction, we hypothesized that tail-chasing could negatively affect the male reproductive success. In our study, however, we have not found any impact of time spent on tail chasing on other observed reproductive traits.

In line with predictions, we have observed increased probability of fertilization success if the contact with females (which we used as a proxy for copulation time, cf. Introduction) lasted longer. This effect held for the ancestral and evolved populations alike and suggests an increased probability of successful sperm transfer for longer matings. We need to emphasize, however, that we did not repeat our experiment and furthermore, the positive correlation between time of contact and fertilization success was a single significant result out of 8 analyses that we performed. Thus, it should be treated as a preliminary result which would require verification in order to be conclusive (*Ioannidis, 2005*; *Moonesinghe, Khoury & Janssens, 2007*). This is important to bear in mind particularly in the light of the current replicability crisis plaguing biological and social sciences (cf. *e.g.*, *Ioannidis, 2005*; *Moonesinghe, Khoury & Janssens, 2007*; *Parker, 2013*; *Baker, 2016*). As a future prospect, comparing the ancestral and evolved generations at the genetic level, which is a subject of an ongoing study, is expected to shed more light on the amount of selection acting on male-biased genes in the obligatory outcrossing populations.

## CONCLUSIONS

Our study did not find support for the major hypothesis that over 97 generations of obligatory outcrossing, males will evolve changes in sexual behaviour and increased efficiency of fertilization. Males after 97 generations were not more successful than their ancestors in finding, keeping contact with, or fertilizing the females. We conclude that selection on male mating behaviour caused by the increased male frequency (1:1) has not been strong enough, or mutational input of new adaptive variants has not been sufficient to cause noticeable behavioural differences after 97 generations in the initially genetically uniform populations. Finally, it is worth emphasizing that our results may be specific to the particular genetic background of the ancestral population used in this study. As discussed above, this may have contributed to their contrasting with similar studies performed on genetically variable populations. Moreover, though, our conclusions should not be extrapolated to genetically uniform populations derived from other backgrounds, which may show different evolutionary responses and/or relationships between traits.

### Funding

The study was financed by the National Science Centre (Poland) grants: 2018/29/N/NZ8/02616 to Weronika Antoł (the behavioural experiment) and 2017/26/E/NZ8/00879 to Zofia M. Prokop (maintaining the experimental evolution). The funders had no role in study design, data collection and analysis, decision to publish, or preparation of the manuscript.

### Grant Disclosures

The following grant information was disclosed by the authors:
National Science Centre (Poland): 2018/29/N/NZ8/02616 and 2017/26/E/NZ8/00879.

### Competing Interests

The authors declare that they have no competing interests.

### Author Contributions

- Weronika Antoł conceived and designed the experiments, performed the experiments, analyzed the data, prepared figures and/or tables, authored or reviewed drafts of the article, and approved the final draft.
- Jagoda Byszko conceived and designed the experiments, performed the experiments, authored or reviewed drafts of the article, and approved the final draft.
- Alicja Dyba conceived and designed the experiments, performed the experiments, authored or reviewed drafts of the article, and approved the final draft.
- Joanna Palka performed the experiments, authored or reviewed drafts of the article, and approved the final draft.
- Wiesław Babik analyzed the data, authored or reviewed drafts of the article, and approved the final draft.

- Zofia Prokop conceived and designed the experiments, analyzed the data, authored or reviewed drafts of the article, and approved the final draft.

## Data Availability

The raw data is available at Figshare: Antoł, Weronika; Byszko, Jagoda; Dyba, Alicja; Palka, Joanna K.; Babik, Wiesław; M. Prokop, Zofia (2022): Dataset to 'Evolution of reproductive behaviour in *Caenorhabditis elegans* under forced obligatory outcrossing'. figshare. Dataset. https://doi.org/10.6084/m9.figshare.20472936.v1.

The code is available at Zenodo: Antoł, Weronika, Byszko, Jagoda, Dyba, Alicja, Palka, Joanna, Babik, Wiesław, & Prokop, Zofia. (2022). Code file for the behavioural experiment. Zenodo. https://doi.org/10.5281/zenodo.7199833.

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
