# Peer review of "No detectable changes in reproductive behaviour of Caenorhabditis elegans males after 97 generations under obligatory outcrossing"

_PeerJ, doi:10.7717/peerj.14572_

## Round 0.1 · original submission · Major Revisions

Dear authors,

I just received the reviews of two reviewers regarding your manuscript entitled "Evolution of reproductive behaviour in Caenorhabditis elegans under forced obligatory outcrossing". You will see that while reviewer 1 has a positive impression of your manuscript (despite pointing out some important issues), reviewer 2 has some serious issues with your experimental design. Since both reviewers had contrasting impressions of your experimental design, I recommend you perform a profound review of the manuscript. In particular, it is important that you address the potential qualities and restrictions of your experimental design, with a particular emphasis on discussing whether the absence of a control population that was allowed to evolve without obligatory outcrossing could affect the robustness of your conclusions.

·

Basic reporting

The manuscript, “Evolution of reproductive behavior in Caenorhabditis elegans under forced obligatory out crossing” reports that serially passaging C. elegans N2 strain that contain the fog-2 (q71) allele for 97 generations does not result in gross alterations to male copulation behavior that can be detected by the authors’ assays. The purpose of their study is to test the idea that obligatory out-crossing might select for behavioral male traits that promote fertility. The authors created 4 lineages of fog-2 (q71) and serially passaged descendants of those lineages for 97 generations. The authors then sampled the copulation behavior of males and their female siblings from generation 1 and generation 97. Their measurements and analyses could not statistically detect any gross behavioral changes between males of the ancestral and 97th generation. The authors conclude that 97 generations was insufficient to evolve copulation fitness traits or the behavioral assays used to measure copulation parameters did not have the precision to detect subtle changes in sensory motor behaviors.

Experimental design

I have no issue with the statistics done in the work or the conclusions made by the authors, but there are some small issues in the writing that can be addressed, which I note in the additional comment section.

Validity of the findings

I am in favor for publishing negative data, and I believe the author’s work is useful for others to know how to alter future experimental design, if others wish to revisit the question of when and how male copulation traits can evolve. The question of what ways male copulation can evolve is of interest to this reviewer, and one of four ways can be envisioned: (a) improvement in sensory-motor coordination used for copulation (which the authors tried to assay here), (b) acquiring traits that promote male attractiveness (Markert et al. 2013. Virgin Caenorhabditis remanei females are attracted to a coital pheromone released by con-specific copulating males Worm 2:3 e24448.) (c) overpowering a potential mate (Garcia et al. 200. Diversity in mating behavior of hermaphroditic and male-female Caenorhabditis nematodes. Genetics 2007; 175:1761-71) and (d) overpowering sibling male competition. This reviewer believes that the authors’ approach using serially passaged fog-2 mutants can get at this question (albeit future experiments probably should not use the q71 allele since that easily reverts back to being a hermaphroditic strain (Katju et al. Sex change by gene conversion in a Caenorhabditis elegans fog-2 mutant. Genetics 2008 180:669-72); however in future experiments, some additional selection needs to be imposed upon the serial passaging paradigm to select for evolved traits. Likely, the authors’ experimental set up does not provide sufficient selection for enhanced copulation alleles to sweep through the population to counter the dilution effects of consistent outcrossing in a artificially constrained environment like a Petri plate.

Additional comments

1. The title, “Evolution of reproductive behavior in Caenorhabditis elegans under forced obligatory out crossing”, needs to be modified since the authors did not detect any evidence of heritable change at the population level (my definition of evolution) in behavior during the 97 generations. Also the word “forced” can be omitted since the word“obligatory” already defines the situation.
2. Line 16, “…of nearly 100 generations of enforced obligatory outcrossing…” reads better as, “… of nearly 100 generations of obligatory outcrossing.”
3. Line 19, “…outcrossing, was split into four independently evolving populations.” Reads better written as “…outcrossing, was split into four independent populations.” There is no sequence (or phenotypic) data provided to the reader of any parameters of evolution occurred (genetic changes that swept through the populations).
4. Typo: line 44 to 45. “Either way, selfing syndrome provides a compelling example of how profoundly can shifts in reproductive systems affect the evolution of morphological, physiological and behavioural traits. Should read as, “ Either way, selfing syndrome provides a compelling example of how profound shifts in reproductive systems affect the evolution of morphological, physiological and behavioural traits.
5. Lines 59-60. “...the latter was suggested to be the main source of males (Chasnov & Chow, 2002). This is a more appropriate citation for this statement: Jonathan Hodgkin, H. Robert Horvitz, and Sydney Brenner Nondisjunction Mutants of the Nematode CAENORHABDITIS ELEGANS. Genetics. 1979 Jan; 91(1): 67–94.
6. Line 64. “…elegans males produce smaller sperm, are less vigorous during mating and rarely, if ever, deposit copulatory plugs on their partner’s vulvas.” The sentence probably needs to be re-written, since the deficiency in copulatory plugging is a problem of just in the laboratory N2 strain of C. elegans. Wild hermaphroditic strains of C. elegans have mating ability and plugging ability that is better than the N2 strain (Hodgkin, J., and T. Doniach, 1997 Natural variation and copulatory plug formation in Caenorhabditis elegans. Genetics 146: 149–164.).
7. Line 67 to 68. “ Also hermaphrodite traits appear to be affected by selfing syndrome: in contrast to females of related species, hermaphrodite C. elegans apparently do not produce the male-attracting pheromone.” This probably needs to be written. They actually do produce an attractive cue (Srinivasan et al. 2008. A blend of small molecules regulates both mating and development in Caenorhabditis elegans. Nature 454. 07168.)
8. Line 76-78 The natural reproductive system of C. elegans can be experimentally changed to dioecy, i.e., obligatory outcrossing, through genetic manipulations (see Table I in Anderson et al., 2010; 78 Gray & Cutter, 2014). Should also include these citations, since they are the primary citations for the statement.
Hodgkin, J., and S. Brenner,1977. Mutations causing transformation of sexual phenotype in the nematode Caenorhabditis elegans. Genetics 86: 275-287.

Hodgkin, J., 1980. More sex determination mutants of Caenorhabditis elegans. Genetics 96: 649-6

Doniach, T., and J. Hodgkins, 1984. A sex-determining gene, fem-1 required for both male and hermaphrodite development in Caenorhabditis elegans. Dev. Biol. 106: 223-235.

Schedl and J. Kimble 1988. fog-2, a Germ-Line-Specific Sex Determination Gene Required for Hermaphrodite Spermatogenesis in Caenorhabditis elegans. Genetics 119: 43-61.

9. Line 116-118. “Our main predictions were that after 97 generations of evolution the males will be able to find the mates more quickly than the ancestral males, maintain physical contact with them for a longer time and be more successful in fertilization.

Should be changed to:
“Our main predictions were that after 97 generations, the males will be able to find the mates more quickly than the ancestral males, maintain physical contact with them for a longer time and be more successful in fertilization.

10. Line 276. “Thus, the difference they observed in spicule insertion time could potentially be resulting directly from the fog-2 mutation itself, as well as from the subsequent evolution.

Should be changed to, “Thus, the difference they observed in spicule insertion time could potentially be resulting directly from the fog-2 mutation itself or a linked mutations in neighboring genes, as well as from the subsequent evolution.”

Base on the biochemical and molecular properties of fog-2, I find it very unlikely that the absence of the fog-2 molecule has a physiological effect on the cholinergic motor system of the male p.c.s and spc sensory-motor neurons or the anal depressor/protractor musculature. More possibly, historical passage of the fog-2 (q71) animals might allowed for accumulation of mating-promoting changes that are linked to the fog-2 region of chromosome; close linkage can make introgression difficult around the fog-2 loci of chromosome V.

11. Line 287- 288. “Our results suggest that 97 generations were not sufficient for such changes to evolve.”

Should be changed to,”

“Our results suggest that 97 generations were not sufficient for such changes to arise.”

Or

“Our results suggest that 97 generations were not sufficient for copulation traits to evolve.”

Reviewer 2 ·

Basic reporting

The manuscript is generally clear. The work is put into appropriate context. The final paragraph of the discussion could be edited for brevity and clarity. There are many ideas introduced in that single paragraph, it would be better to streamline it a bit so that the primary questions of the work are clear.

Experimental design

I have serious concerns about the experimental design of this study, assuming I correctly understand the experimental design and motivation of this work.

The work is presented as a project that contrasts obligate outcrossing in the experimental populations with the previously mixed mating populations of C. elegans. However, there are no controls. To make this comparison appropriately the mixed mating populations that served as the progenitor for the obligately outcrossing population would need to be evolved alongside the obligately outcrossing populations for 97 generations. Then, comparisons could be made between the males after evolution. As it stands, there is no point of comparison apart from the ancestor. Comparisons to the ancestor are good for observing change. However, comparing only to the ancestor does not permit researchers to make conclusions about how or why that change occurred. If evolved, the mixed mating populations may have produced the same changes, thus the changes observed here would not be attributable to obligate outcrossing. Without a control, there is no way to know how the mixed mating populations would have responded. Further, it is unclear if the methods of experimental passage were the driver of evolutionary change, as opposed to obligate outcrossing itself. Ultimately, this work is an observation of the evolutionary trajectories of 4 populations, but it is not an experiment that can be generalized beyond those 4 populations in this specific set of conditions.

Validity of the findings

Due to the issues with experimental design, the conclusions of the manuscript are not supported by the data. Without controls, no general conclusions can be valid.

Additional comments

The basic premise of the work is quite intriguing. I believe testing the evolutionary trajectories of obligate outcrossing vs mixed mating populations of C. elegans would be very interesting, particularly in relation to Palopoli et al’s results. However, appropriate controls are necessary to make it a viable experiment. I highly recommend conducting such an experiment.

---

## Round 0.2 · Minor Revisions

Dear authors,

First, I would like to apologize for the delay to review the manuscript. Due to the very divergent reviews of the first version, I decided to ask for an additional review in this round. You will see that while reviewer 1 was very positive about the manuscript and has no additional suggestions, the new reviewer has some minor concerns. For this reason, I ask you to check the comments from reviewer 2 before I make a final decision.

Best wishes,

·

Basic reporting

The authors addressed my suggestions.

Experimental design

The authors addressed my suggestions

Validity of the findings

The authors addressed my suggestions

Additional comments

Good Job.

Reviewer 3 ·

Basic reporting

The paper is well written, and the English is clear.

The Introduction is relatively thorough and the citations well covered. However, in both the Introduction and Discussion it would be briefly reflecting on the use of N2 as the strain background in the study. After decades of use N2 is well known to have acquired many mutations that lead to its lab adaptation. Additionally, traits in males such as ability to create cross progeny (mating frequency) and sperm size have been shown to be less robust in N2 versus other C. elegans isolates- see Bahrami and Zhang et al. 2013 and Gimond et al., 2019. Given the difference seen here versus previous similar experiments that used other genetic backgrounds this could be at play in this work.

The figures are generally clear and well built. The methods figures are especially helpful. However, in Figure 3f-h it is not clear why the dots associated with a particular strain do not overlap with the box plot for that strain. This makes it very difficult to interpret the spread of the data within each dataset. Two examples, first: in g in the “1” category the two dots that represent the number of tail chases in the ancestral strain are over the box plot for the strains that evolved for 97 generations; second: in h for the “K05” data we know from the text that only one male produced cross progeny – the dot that represents that male’s contact duration I assume is the dot to the left of the line that represents the “1” and then there are a number of dots below that line that instead represent the contact duration for males that did not produce cross progeny. It would be much clearer and less work for the reader if the dots associated with a particular box and whisker plot overlapped like in graphs a,b, d.

Experimental design

The experiments are well explained and would be replicatable.

The authors do a good job of laying out the specific reasoning behind their set-up: specifically using an inbred line to mimic small population size and looking at accumulation of genetic change on phenotypic outcome. The only place that is somewhat unclear is when males are isolated before the experiment on Day 1 if they are placed of food similar to females or not.

Validity of the findings

The authors present and interesting test of an interesting question. They also clearly state that their findings should be viewed as preliminary as they haven’t been replicated and carefully interpret their results due to the constraints of their methodology. It is clear why replicating the entire experiment would require a large investment of time and resources- but it isn’t clear why the experiments have such a small N (between 9 and 10 worms tested).

---

## Round 0.3 · accepted · Accept

I have carefully reviewed the last version and the authors were able to satisfactorily answer all comments from reviewer 3. For this reason, it is my pleasure to recommend acceptance. Below, I indicate just two small typos that I found in the text.

Line 65: provide a space between "mating" and "(Chasnov et al., 2007...". Also insert a period at the end of this sentence.